# The Water Flux Dynamic in a Hybrid Forward Osmosis-Membrane Distillation for Produced Water Treatment

**DOI:** 10.3390/membranes10090225

**Published:** 2020-09-09

**Authors:** Normi Izati Mat Nawi, Muhammad Roil Bilad, Ganeswaran Anath, Nik Abdul Hadi Nordin, Jundika Candra Kurnia, Yusuf Wibisono, Nasrul Arahman

**Affiliations:** 1Department of Chemical Engineering, Universiti Teknologi PETRONAS, Bandar Seri Iskandar, Perak 32610, Malaysia; normi_16000457@utp.edu.my (N.I.M.N.); neshhh09@gmail.com (G.A.); nahadi.sapiaa@utp.edu.my (N.A.H.N.); 2Department of Mechanical Engineering, Universiti Teknologi PETRONAS, Bandar Seri Iskandar 32610, Perak Darul Ridzuan, Malaysia; jundika.kurnia@utp.edu.my; 3Bioprocess Engineering, Brawijaya University, Malang 65141, Indonesia; y_wibisono@ub.ac.id; 4Department of Chemical Engineering, Universitas Syiah Kuala, Banda Aceh 23111, Indonesia; nasrular@unsyiah.ac.id

**Keywords:** membrane distillation, forward osmosis, produced water treatment, process integration, membrane-wetting

## Abstract

Standalone membrane distillation (MD) and forward osmosis (FO) have been considered as promising technologies for produced water treatment. However, standalone MD is still vulnerable to membrane-wetting and scaling problems, while the standalone FO is energy-intensive, since it requires the recovery of the draw solution (DS). Thus, the idea of coupling FO and MD is proposed as a promising combination in which the MD facilitate DS recovery for FO—and FO acts as pretreatment to enhance fouling and wetting-resistance of the MD. This study was therefore conducted to investigate the effect of DS temperature on the dynamic of water flux of a hybrid FO–MD. First, the effect of the DS temperature on the standalone FO and MD was evaluated. Later, the flux dynamics of both units were evaluated when the FO and DS recovery (via MD) was run simultaneously. Results show that an increase in the temperature difference (from 20 to 60 °C) resulted in an increase of the FO and MD fluxes from 11.17 ± 3.85 to 30.17 ± 5.51 L m^−2^ h^−1^, and from 0.5 ± 0.75 to 16.08 L m^−2^ h^−1^, respectively. For the hybrid FO–MD, either MD or FO could act as the limiting process that dictates the equilibrium flux. Both the concentration and the temperature of DS affected the flux dynamic. When the FO flux was higher than MD flux, DS was diluted, and its temperature decreased; both then lowered the FO flux until reaching an equilibrium (equal FO and MD flux). When FO flux was lower than MD flux, the DS was concentrated which increased the FO flux until reaching the equilibrium. The overall results suggest the importance of temperature and concentration of solutes in the DS in affecting the water flux dynamic hybrid process.

## 1. Introduction

Membrane distillation (MD) is a thermally driven process that transport water molecules in vapor phase through porous and hydrophobic membrane pores. Theoretically, MD requires lower operating temperatures compared to the conventional distillation process with similar total rejection of macromolecules, colloids, cells, ions and other non-volatiles. In addition, MD suffers less concentration polarization than pressure-driven membrane processes such as RO and is thus suitable to treat high solute feed [1]. Such advantages lead to recent explorations of MD application for produced water treatment. However, membrane fouling, temperature polarization and pores wetting are some of the major challenges for MD process that would affect its durability and performance if it were used as a standalone unit [2,3], when run without pretreatment especially when treating feed containing hydrophobic droplets such as produced water.

Although a total rejection of solute is expected in MD process for non-volatile solutes such as NaCl, the rejection is often less due to the membrane defect or membrane pore wetting [4,5]. Membrane pore wetting is caused by the penetration of feed solution into the membrane pores, block the pore passage and reduce the available pores for vapor transportation thus deteriorating the flux performance. It is worth to note that partial wetting can be easily detected by a decrease of water flux while total wetting contributes to sudden increase of flux but depresses its rejection performance during MD operation. Membrane-wetting can also be caused by the direct contact of feeds containing wetting compounds such as oils and surfactants—such as produced water-making application of standalone MD application quite challenging [6]. To some extent, membrane degradation would occur due to the formation of hydrophilic chemical group with long term use of membrane which disturb its chemical and mechanical stability [7]. Therefore, integrating FO with membrane distillation (MD) has been widely explored [8].

Forward osmosis (FO) process has been extensively investigated for a wide range of applications since 1930’s [9]. In contradictory to reverse osmosis (RO) process, FO uses osmotic difference to drives the mass transport across a semipermeable membrane. The whole process requires a high salinity draw solution (DS) and low salinity of feed solution (FS) separated by the FO membrane. The difference of osmotic pressure across the membrane drives the mass transport of water from the feed side to the draw side and simultaneously reverse salt transport in the opposite direction [10,11,12]. Since the osmotic process occurs naturally without any additional pressure applied on the system except for the pumping of FS and DS, it offers higher total dissolved solids (TDS), high salt rejection and FO has been widely cited as a low or a highly reversible fouling process [11,12,13].

Despite offering numerous advantages than the pressure driven membranes such as RO, the FO water flux is highly dependent on the osmotic pressure generated from the DS [11]. In batch process with the circulation of the DS and the FS, the rate of liquid water transports across FO membrane declines over time because of the dilution of DS and concentration of FS [14]. At certain point, it eventually reaches an equilibrium osmotic pressure, which leads to zero net flux [15]. In most cases, FO can be viewed as a pretreatment process because the process is not completed in single step and most of the time cannot be run as a process especially when abundant source of concentrated solute (such as seawater) is unavailable [16,17]. This is why DS requires additional step of DS regeneration/concentration to form an integrated process. This drawbacks possesses serious challenge for scaling-up of FO since a standalone process (just like RO or membrane distillation, MD) is (theoretically) more preferred [10,18]. The necessity of including additional process thus raise a question of whether a combination of two-processes offer better efficiency than a single standalone process (i.e., FO+RO vs. standalone RO), because more processes lump into lower overall efficiencies.

Most studies treat FO as an integrated process and thus coupled with a DS recovery system (such as RO, MD or others [19]) [10], but they focus less on the technical impact of the hybridization. For instance, Aydiner et al. [20] integrated FO with RO for whey concentration and water recovery. The experiments of FO and RO were conducted separately and thus could not capture the dynamics of the process integration. Nonetheless, they reported the detrimental effect of reverse salt flux to the concentrated whey as the FS and some soluble organics being permeated into the DS. Meanwhile, Kumar and Pal [21] integrated FO with NF (nanofiltration) for coke-oven wastewater treatment and run the tests as standalone FO, standalone NF and combination of both in an integrated system. A reasonably high flux of 46 LMH was achieved in long-term operation. When running the integrated FO-NF, they found the decreasing trend of flux for both FO and NF over time. However, the dynamic of the hybridization was not obvious and not fully addressed from the trend, because the FO and fluxes seems to be independent. Thus, the concept of recirculating diluted DS from FO process in FO–MD integrated system to concentrate the solution, is found interesting for wide range of applications [22,23,24,25]—particularly in addressing the fouling and wetting issue of MD membrane during long-term operation. In the context of treating produced water, the heat source to run the MD can be obtained by recovering the excessive heat released via the flare to ensure the sustainability of the treatment process.

The FO unit in the hybrid FO–MD system is expected to serve as a low-cost and high-efficiency pretreatment for MD to remove the substances from feed solution that potentially cause wetting while offering a great solution to membrane fouling in MD [26], since MD is only involve in treating sole draw solutes from FO [27]. At the same time, MD also shows great potential for small-scale wastewater reclamation, decentralized systems with available waste heat source [2], as well as for waste heat recovery from the flue gas [28].

Given the promising potential of FO and MD integration, numerous studies exploring optimum combination to take advantages of both processes for numerous applications have been conducted and reported, some of which are summarized in Table 1. Wang et al. [29] concentrated protein solution from 1 g L^−1^ to roughly 2.1 g L^−1^ in a 4 h operation. A high-water flux of 25 L m^−2^ h^−1^ was obtained by Ge et al. [30] when treating dye containing wastewater. Liu et al. [31] recorded nearly complete rejection of total organic carbon (TOC), total nitrogen (TN) and ammonium nitrogen (NH_4_^+^-N) regardless of DS concentration for human urine treatment. Meanwhile, Lee et al. [32] developed a FO–MD hybrid system to treat flue gas desulfurization wastewater obtained from a coal-fired power plant in Korea. High removal rates of 98.61%, 99.97% and 99.93% for organic, ions and particles were achieved using the integrated system. The hybrid FO–MD system also has been used to treat real dairy wastewater and reported almost total rejection of TOC (99.86%), TN (99.31%) and TP(100%) [33]. Nonetheless, for all those reports, the FO–MD system were run as standalone FO or standalone MD processes, without putting emphases on the dynamic of process intensification when run simultaneously.

Several reports on simultaneous operation of FO and MD in a hybrid process were reported. Li et al. [27] reported a good separation performance of an integrated FO–MD system for wastewater treatment and almost total rejection for most contaminant asides from satisfying performance of flux stability. Hybrid FO–MD system by Xie et al. [25] also demonstrated 80% water recovery and excellent removal of trace organic contaminants with removal rates ranging from 91% to 98% to treat raw sewage. The effectiveness of FO as pretreatment to limit fouling in MD and to reduce the membrane-wetting vulnerability has also been reported. However, the dynamic of the FO and MD fluxes overtime during the hybrid test was not revealed. Despite the emerging efforts on understand the FO–MD hybrid system, detailed investigation on the dynamic of the hybridization as one-unit module is limited. Thus, more studies are necessary. The integration of FO and MD facilitates mutual advantages: MD facilitate DS recovery for FO, and FO acts as pretreatment to enhance fouling and wetting-resistance of the MD. This study investigates the performance of the FO–MD hybrid system, particularly on the effect of DS temperature on the dynamic of the fluxes between the two units, for treating oily wastewater.

## 2. Materials and Methods

### 2.1. Membrane and Materials

A thin film composite membrane with hydrophilic polyamide layer (TFC-PA; Toray, Tokyo, Japan) was used for FO, while a PTFE membrane Fluoropore^®^ (Millipore, Burlington, NJ, USA) was used for the MD. Both flat sheet membranes had the same effective area of 3.4 × 10^−3^ m^2^ attached to the respective FO or MD filtration cells. The TFC-PA membrane employed for FO module had a thickness of 108.6 µm with a thin dense polyamide layer acts as active layer to ensure high solute rejection. The membrane showed hydrophilic surface properties with the contact angle value of 52.85° as reported elsewhere [37]. The pure water permeability coefficient, the solute permeability coefficient and the structural parameter of the applied membrane were 5.36 L m^−2^ h^−1^ bar^−1^, 0.95 L m^−2^ h^−1^ and 0.265 mm, respectively as reported elsewhere [38]. In contrast, the PTFE membrane used for MD operation possess hydrophobic properties of 138° contact angle, pore size of 0.22 µm, thickness of 175 µm and 70% of membrane porosity as reported in our earlier report [28].

### 2.2. Properties of Feed and Draw Solution

A real PW (produced water) was obtained from a local petroleum company. It was used as the feed for all the filtration tests. The DS for the FO was prepared from 0.6 M of NaCl solution to mimic the seawater. Both PW and DS were characterized in term of conductivity, total organic carbon (TOC), turbidity and emulsified oil content using a conductivity meter (Hanna Instrument, Woonsocket, RI, USA), Hach-Lange kits (Hatch, Colorado, CO, USA), TOC analyzer (VCSH, Kyoto, Japan), turbidity meter (2100Q, Hach, Loveland, CO, USA) and UV-Vis Spectrometer (DR5000, Hach, Loveland, CO, USA). It was found that the conductivity of PW and DS were 2000 and 40,152 μS/cm measured at room temperature of 20 °C, respectively. Such high osmotic pressure of DS (seawater) is desirable for FO to ensure enough driving force to facilitate water mass transport across the membrane. The PW feed posed 583 of TOC mg L^−1^, 79.4 NTU of turbidity and contained 88.43 mg L^−1^ emulsified oil. Meanwhile, the DS had 8.86 mg L^−1^ of TOC and 0.91 NTU of turbidity without emulsified oil content. The rejection performance of the standalone FO system can be obtained in our earlier report [37]. Analysis of the MD permeate showed that the system fully retained the TOC and turbidity.

### 2.3. Filtration Setup

A custom-made hybrid lab-scaled FO–MD system was designed to conduct the standalone FO and MD and the hybrid FO–MD system tests as schematically shown in Figure 1. For the FO process, it consisted of a flat sheet membrane cell, feed and draw solution tanks, weighing balance and peristaltic pumps to recirculate the fluid in separate enclosed loops. The changes of weight of the feed tank containing PW were continuously monitored over the filtration test using a weighing balance for the FO water flux calculation. The system was equipped with a heater in the DS circulation to control the temperature of the DS. For MD process, it had almost similar setup to the FO except that the changes of weight of the permeate tank was continuously monitored for the MD water flux calculation to continuously monitor the MD flux. Direct contact membrane distillation configuration was applied for the MD system. DI water was used as the stream on the cold side of permeate. The FO and the MD system were first run independently to study the effect of temperature difference on the performance of each system. For each experiment, the initial volumes of the FS and the DS were set at 500 and 300 mL, respectively.

To study the effect of DS temperature on the FO and MD flux, the temperature of DS was set at 30, 40, 50, 60, 70 °C which corresponds to temperature differences of 20, 30, 40, 50 and 60 °C, respectively. The PW feed and MD distillate were maintained at 10 °C for the whole experiment and each test was run for 120 min. The used membrane was replaced with a new one after each set of experiment. The reported flux values were obtained the average of at least three readings. The fluxes for both FO and MD operations were calculated by using Equation (1) and the flux average of at least three readings was determined. Equation (1) was as follows:(1)$$J_{W}=\frac{∆V}{A∆t}$$
where *J_W_* is permeate flux (L/m^2^ h or LMH), *A* is effective membrane area (3.6 × 10^−3^ m^2^), ∆*V* is volume of permeate collected (*L*), ∆*t* is filtration period (*h*). The system circulated the diluted DS from the FO unit into the hot side of MD unit directly for condensation.

After evaluating the individual FO and MD system, the performance of the integrated FO–MD mode was then assessed to investigate the correlation of ∆*t* on the flux. The DS was first set at 50 °C, corresponds to the ∆*t* of 40 °C and then increased to 70 °C, corresponds to ∆*t* of 60 °C to investigate the flux performance at higher ∆*t* and it was conducted for 80 min. The two ∆*t* were selected to demonstrate the dynamic of the integrated FO–MD module in response to two values of driving forces. The operation was extended for 270 min for the ∆*t* = 60 °C to study the fluxes dynamic of the integrated FO–MD system.

## 3. Results and Discussion

### 3.1. Effect of Draw Solution Temperature on the Standalone FO Water Flux

Figure 2 shows the performance of FO at varied DS temperatures. It shows that the FO flux increased from 11.17 ± 3.85 to 30.19 ± 5.51 L m^−2^ h^−1^ when the temperature differences was increased from 20 to 60 °C, which was comparable with other studies [36,39,40,41]. As reported by Kim et al. [36], the significant augmentation of water flux could be justified by analyzing two effects: (*i*) an increase in osmotic pressure induced by high temperature, according to the van ’t Hoff equation [42] and (*ii*) an increase in the diffusivity of draw solutes due to an increase of DS temperature according to Stokes–Einstein equation which reduced the mass transfer resistance [43]. Increasing the temperature of DS promoted the heat transfer to the FS across the FO membrane due to temperature polarization. This also lead to a decrease of internal concentration polarization and significantly elevated the effective concentration gradient across the active layer, thus improve the process driving force [39,44]. The results obtained are desirable for the integrated FO–MD system since the higher temperature DS is circulated in the hot feed stream of the MD process. However, excessive loss of heat maybe detrimental when run in full-scale setup—and as demonstrated later—affect the flux dynamic in the FO–MD hybrid process.

### 3.2. Effect of Feed Temperature on the Standalone MD Water Flux

As explained previously, MD is a thermally driven separation process. Therefore, temperature difference was expected to have a dominant effect to its flux performance, as demonstrated in Figure 3 where the permeate flux was very sensitive to the feed temperature. A higher flux was obtained at higher feed temperature. For a FS temperature of 20 °C, only 0.5 ± 0.75 L m^−2^ h^−1^ of water flux was attained, and the MD process achieved a high flux of 16.08 ± 4.90 L m^−2^ h^−1^ when the FS temperature was increased to 70 °C. Such increment was expected since temperature different was the main driving force in an MD. Similar trends of flux performance has been reported by Nawi et al. [1] where the fluxes increased with increased feed temperature from 45 to 65 °C, due to the increase of partial pressure difference which enhance mass transportation.

This temperature dependent flux enhancement can be explained by vapor pressure vs. temperature relationship. An increase of temperature difference exponentially elevated the difference in water vapor pressure across the membrane, which was the main driving force for MD [1,45,46] thus improves the mass transfer coefficient. According to Ge et al. [47], the feed temperatures also influence the viscosity, Reynold’s number, feed hydraulic, temperature polarization coefficient and MD coefficient. Increasing the feed temperature reduces feed viscosity and may cause a thinning of the feed boundary layer as well as an improvement of heat transfer efficiency that leads to low temperature polarization. Temperature polarization caused the boundary layers of both feed and permeate sides to differ from the bulk temperatures [48]. Since direct contact MD configuration was applied, temperature polarization was severe due to the contact of hot feed side and cold permeate side to the membrane surface. However, its flux performance improved by increasing the feed temperature to reduce the effect of temperature polarization, as proven by this study as well as others [49,50,51,52]. Despite of the high flux obtained with the employment of high feed temperature, it is important to note that increasing feed temperature could also increase the risk of scaling and membrane-wetting attributed by oversaturation in the boundary layer for the high concentration feed solutions [47].

### 3.3. Effect of Temperature Difference on the Hybrid FO–MD Module Performance

Figure 4 demonstrates the water flux of the hybrid FO–MD system in treating PW at varied DS temperature. While maintaining the feed solution and distillate at 10 °C, the DS was heated and maintained at 50 and 70 °C, which corresponds to temperature difference of 40 °C (see Figure 4a) and 60 °C (see Figure 4b). It is worth noting that since both FO and MD were operated as an integrated system, the overall hydraulic performance of the system at the equilibrium was then dictated by the lower performing standalone process.

FO offers superior flux performances compared to MD at earlier stage of filtration, until becomes similar at some point of filtration time, as indicated by Figure 4a. The first reading of the FO and the MD water fluxes (after 20 min of filtration) were 20.8 and 4.2 L m^−2^ h^−1^, respectively which equal to almost 80% flux difference. However, the gap of the flux difference was reduced with the filtration time until both FO and MD show similar flux at 8.3 L m^−2^ h^−1^ after 60 min due to the deterioration of FO flux and increment of MD flux. Wang et al. [29] reported that both FO and MD process achieved stable rate of water transfer after 60 min of operation when DS was about 57 °C. The decline of FO flux in this study was as expected since the high FO water flux was not balanced by the MD water flux leading to accumulation of water in the DS. Water accumulation then lead to low DS concentration over time, which diminished the FO water flux. In addition, since the experiments were carried-out batch-wise, the FO feed volume lowered over time and thus the concentration of solutes in the feed solution increased accordingly. Due to the occurrence of reverse salt flux, ions transferred from the draw solution to the feed solution which lowered the osmotic pressure difference between the 2 sided of the FO membranes, thus reducing the driving force. The effect of the reverse salt flux and feed concentration was significant because of the high reverse salt flux of the FO membrane as well as high volumetric reduction ratio (VRR, which reached 14.4% by the end of the experiment).

The slight increase in MD water flux can be explained by the dilution effect of the DS coupled with a slight increase in temperature difference as shown in Figure 4a. After 40 min of filtration time, a slight increase on the temperature difference (from 37.7 to 39.4) was observed. Moreover, the dilution of the DS (feed for the MD) enhanced the mass transfer and enhanced its flux performance. This was because the diluted DS/MD feed developed higher water vapor pressure that enhance the mass transfer across the MD membrane [31]. The same trend of flux performance against time was obtained when the temperature difference was fixed at about 60 °C—except for the meeting point of fluxes, which occurred at about 20 L m^−2^ h^−1^ at around 78 min of filtration time. The delayed flux meeting point was most likely due to the unstable fluxes of both FO and MD and a large discrepancy of the initial water flux value.

The dynamic of batch-wise integration of an FO–MD module is shown in Figure 5. Data in Figure 4 shows that the overall performance of the integrated system was controlled by the MD, the performance limiting unit. Therefore, an additional experiment was performed to investigate the system dynamic, as well as to demonstrate the situation where FO may act as the limiting unit. The situation where FO acted as the limiting stage was possible because of the dilution effect of the DS. Initially, MD unit was the limiting shown by higher water transfer rate in the FO cell than in the MD cell during the first one hour of the test. However, FO then turned to be the rate limiting step when the filtration was further extended. The dynamic of performance overtime was expected because of the unsteady state nature of the process until reaching the system equilibrium. Under equilibrium condition, the water fluxes of both FO and MD were equal.

Figure 5 shows that the limiting factor for the hybrid FO–MD system changed over time. The MD was the limiting unit for the hybrid system at early stage of filtration (the first one hour), but it was later replaced by the FO unit until the end of the test (up until 260 min). The trend of water transfer rates of this work is in line with a study by Li et al. [27]. They reported a reduction of FO flux during 120 h of filtration operation attributed to membrane fouling that deteriorated the FO flux. In contrary, the decline of the flux in the present study was most likely due to excessive dilution of the DS and partly due to decrease in DS temperature. The dilution of the DS lead to gradual decrease of concentration gradient (the driving force of transport in FO) until reaching the equilibrium (equal flux of FO and MD).

The factors affecting flux dynamic after the first equilibrium are somewhat complex. They included concentration of the DS and a change in temperature of the FS and DS over time. After the first flux equilibrium, the FO flux was lower than the MD flux. Under this condition, the solute concentration of the DS increased—by assuming of low reverse salt flux in FO and complete rejection of solute by the MD. The reverse salt flux was calculated as 2.35 g m^−2^ h^−1^ on the standalone FO system. Logically, such situation promotes higher FO flux. However, the results show otherwise. We suspect the further decrease in the FO flux was due to dynamic of the DS and FS temperature. As the DS was diluted, its temperature was also decreased (the temperature different in MD was maintained by lowering the cold side temperature). At the same time, heat loss to the FO FS also contributed to lower the DS temperature. Those results in a steep declines of FO flux, as it was proven in Figure 2 that it was highly sensitive to the DS temperature. The effect of DS temperature was more prominence than the effect of the concentration of the DS. Eventually, the FO flux reaches a minimum value at 150 min of operation, from which it increased until reaching the second flux equilibrium. Another important factor that dictate the flux dynamic was the VRR, particularly as a result of batch-wise operation. As the water permeated from the FO FS, the feed volumes decreased over time, leading to higher concentration of the DS. The compounding nature of the VRR and the reverse salt flux leads to a lower FO flux. It involved mainly in the initial stage of the test where the FO flux decreased steeply. By the end of the test, the VRR reached 67.8%.

The increase in the FO flux at 150 min of operation can be attributed to more dominant impact of DS concentration than the DS temperature. From the first flux equilibrium to this point (at 150 min), the situation where MD flux was higher than the MD flux resulting in the increase of solute concentration in the DS, favoring higher FO flux. This condition proceeded until the second flux equilibrium was achieved, after which further flux dynamic may continue around the equilibrium point. This finding demonstrated the important of DS temperature in dictating the performance of hybrid FO–MD system.

The issue of membrane-wetting in MD was not observed during the entire tests. Typically, it can be observed from the declines of the water flux over an extended operation. This finding demonstrates mutual advantages of FO–MD combination, in which the FO acted as pretreatment of MD which otherwise vulnerable to fouling by hydrophobic materials such as oil droplets present in the produced water. Nonetheless, despite obtaining excellent distillate quality, accumulation of non-volatile contents from the feed still occurred either in the feed concentrate or in the draw solution. Eventually their accumulation may also affect the MD permeate quality, especially when membrane-wetting occurs. At the same time, MD process continuously reconcentrates the diluted DS for the FO process while producing high purity permeate. In addition, the freedom to select the type and concentration of the solute in the DS—feed of MD—also contributes positively to reduces membrane-wetting vulnerability in MD.

## 4. Conclusions

This study investigates the effect of DS temperature on the standalone FO, standalone MD and the flux dynamic of the hybrid FO–MD system when run as an integrated system. The overall results suggest the importance of temperature and solute concentration of the DS in affecting the hybrid process. Results show that DS temperatures strongly affects the standalone MD and FO fluxes. The impact was more pronounced to the FO flux than to the MD flux—despite that water transport in MD was driven by the temperature difference. For the FO, the flux increased from 11.17 ± 3.85 to 30.19 ± 5.51 L m^−2^ h^−1^; while for the MD, the flux increased from 0.5 ± 0.75 L m^−2^ h^−1^ to 16.08 ± 4.90 L m^−2^ h^−1^ when the FS temperature increased from 20 to 60 °C. The difference in flux sensitivity toward the temperature affected the flux dynamic when run under FO–MD hybrid system. Both MD and FO could act as the limiting process, which then dictated the equilibrium flux. Apart from the DS concentration, the DS temperature played important role in affecting the flux dynamic. When the FO flux was greater than the MD flux, the DS was diluted which then lowered the FO flux until reaching an equilibrium flux. Under prolonged dilution, the DS temperature decreased, which further lowered the FO flux. When the FO acted as the limiting unit (FO flux < MD flux), the FO flux increased when substantially high-solute concentration in the DS was achieved. It increased the FO flux to reach the flux equilibrium (FO flux = MD flux). In the FO–MD system, heating of small volume of the DS may ease the technicality in recovery of low-grade heat as an energy source to drive the process.

## Figures and Tables

**Figure 1 membranes-10-00225-f001:**
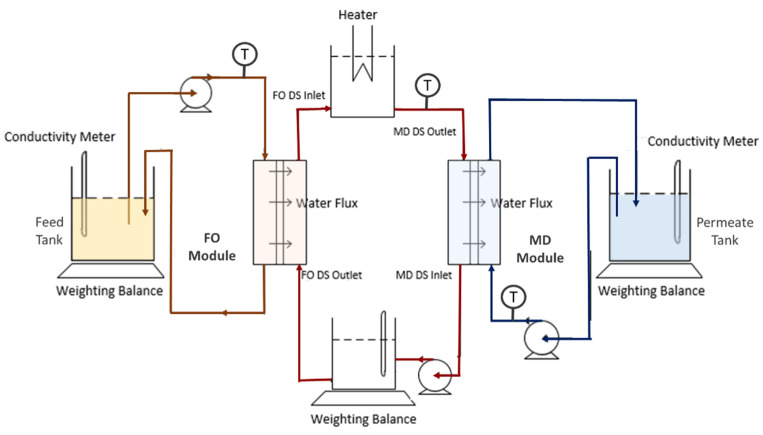
Schematic diagram of a FO–MD hybrid system operated with separated loops.

**Figure 2 membranes-10-00225-f002:**
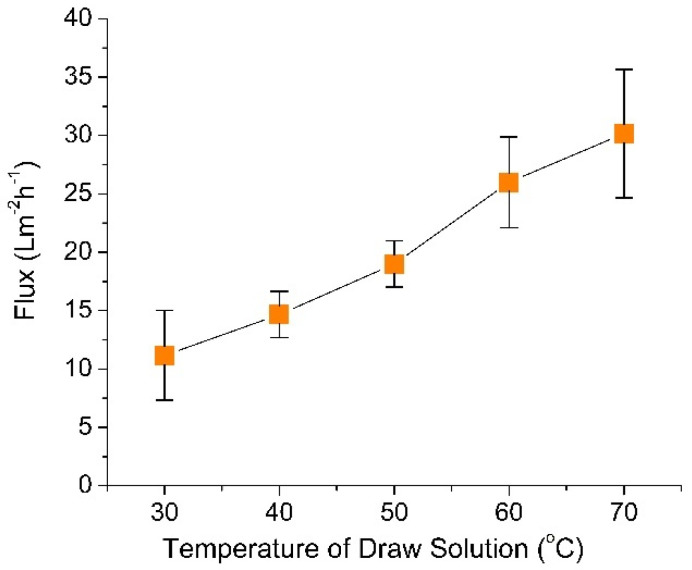
Effect of draw solution temperature on the flux performance of FO. Feed solution was maintained at 10 °C.

**Figure 3 membranes-10-00225-f003:**
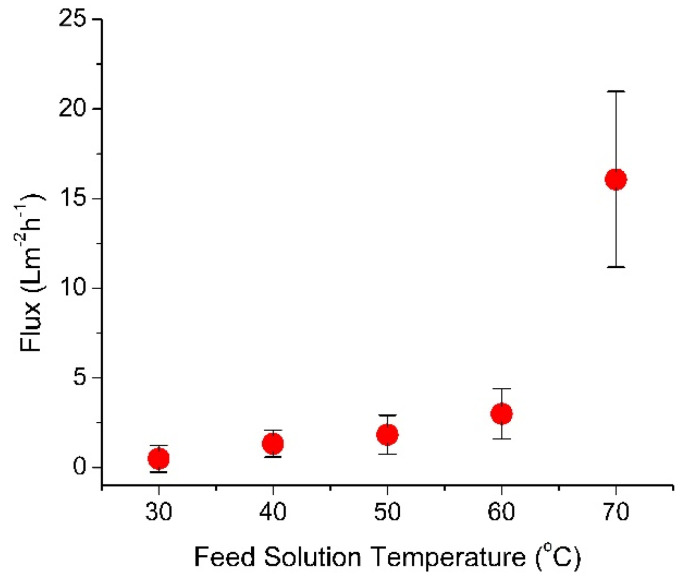
Effect of feed temperature on MD performance.

**Figure 4 membranes-10-00225-f004:**
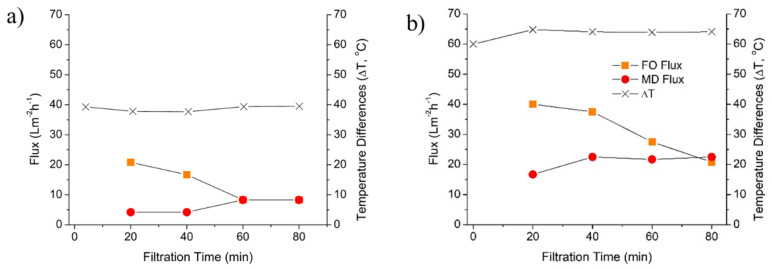
Flux performance of the hybrid FO–MD at temperature differences of (**a**) 40 °C and (**b**) 60 °C with respect to filtration time.

**Figure 5 membranes-10-00225-f005:**
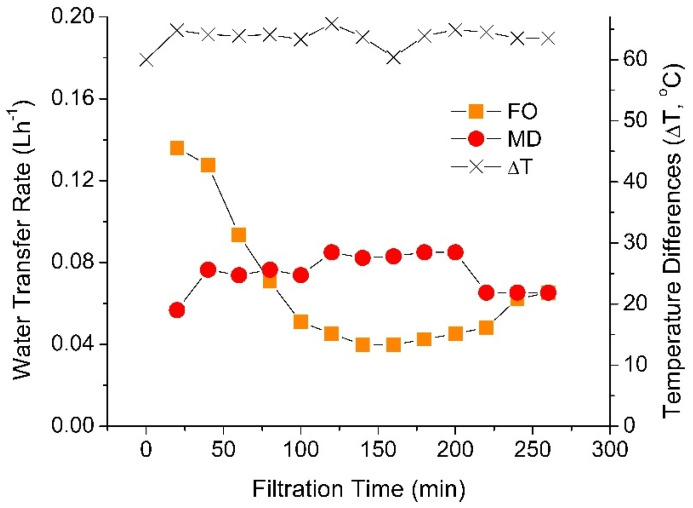
Water transfer rate of FO and MD of 260 min of extended operation performed at 60 °C temperature difference.

**Table 1 membranes-10-00225-t001:** Recent studies of hybrid forward osmosis (FO)–membrane distillation (MD) process for separation of various feed.

Feed	Draw Solution	Membranes	Performances	Remarks	Ref.
Protein solution	NaCl (0.5–2.0 M)	FO: PBI MD: PVDF–PTFE	Feed solution can be concentrated from 1-g/L to roughly 2.1-g/L in 4 h	Water transfer rate of FO–MD balance at 50 to 60 °C	[29]
Dyed wastewater	Poly (acrylic acid) sodium (0.2–0.5 M)	FO: CA MD: PVDF	Highest FO and MD fluxes is 32.0 and 25.0 LMH, respectively, achieved at 80 °C	Water transfer rate of FO–MD most stable at 66 °C	[30]
Synthetic wastewater	NaCl (0.5–2.0 M)	FO: CA–propionate MD: PVDF	Highest FO and MD fluxes is 19.9 and 16.2 LMH, respectively, achieved at 70 °C	Most similar flux at 0.5-g/L feed concentration	[24]
Landfill leachate	NaCl (optimal: 4.8 M)	FO: TFC MD: PTFE–PVDF	Highest FO and MD fluxes is 4.3 and 6.2 LMH, respectively	Water transfer rate of FO and MD stable at 62.5 °C (100,000-mg/L NaCl in feed solution)	[34]
Domestic wastewater	NaCl (0.6 M)	FO: CTA MD: PVDF	Water flux maintained at 17.6 LHM for 120 h operation	After 120 h operation, fouling on FO membrane reduces the flux	[27]
Produced water	NaCl (5 M) KCl (4 M) MgCl_2_ (4 and 4.8 M) LiCl (4.8 and 10 M)	FO: HTI–TFC MD: 3 M–PP	Stable fluxes (3.0–4.0 LMH) for 20 h operation achieved at 4.8-M MgCl_2_ of draw solution	NaCl, KCl, LiCl show high MD flux and low or negative FO flux	[35]
DI water	MgCl_2_ (0.37, 0.62 and 1.44 M)	FO: CTA MD: PTFE	FO flux 13.0 LMH at 50 °C and MD flux 102 LMH at temperature difference of 60 °C	Initial flowrate and concentration are the most important factors for stable integrated module	[36]

NaCl—sodium chloride; PBI—polybenzimidazole; PVDF—polyvinylidene(fluoride); PTFE—polytetrafluoroethylene; CA—cellulose acetate; CTA—cellulose triacetate; TFC—thin film composite—DI—deionized; LMH—L m^−2^ h^−1^.

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
