# Peer review of "The Water Flux Dynamic in a Hybrid Forward Osmosis-Membrane Distillation for Produced Water Treatment"

_membranes, 2020, doi:10.3390/membranes10090225_

Round 1

Reviewer 1 Report

In this manuscript, the authors presented work on the combination of the advantages of FO and MD, which enhance fouling and wetting resistance of the MD. This paper focus on the temperature impact of the FO and MD hybridization system to flux dynamic, which complements the certain research vacancy. After evaluating the overall topic of the work, which seems to address an important of the field, however, major revision need to be conducted before accepted in this journal.

  1. Language usage should be improved and paragraph logic should be stated more clearly. For example, in the 2nd paragraph of Introduction, the sentence “It can be easily detected…” does not function as logical connection between the preceding and the following part. The first sentence in the 3rd paragraph of Introduction is constituent redundancy since the relevant semantics have been described above.
  2. How many samples were tested should be written in Experimental part, as it is clear that reliability and repeatability are important.
  3. This work just focus on the temperature impact of the FO and MD hybridization system to flux dynamic, but it is insufficient in the depth of research. This reviewer think the author should conduct in-depth research on relevant issues, such as the function relationship between the temperature and the flux dynamic.
  4. Last but not the least, please improve the discussion. In many places of the text, there are only description of the experimental results, but no discussion on the reason, e.g. why could the temperature have impact on the flux dynamic?

Reviewer 2 Report

The paper “The Water Flux Dynamic in a Hybrid Forward Osmosis-Membrane Distillation for Produced Water Treatment” describes the dynamic behavior of a batch hybrid forward osmosis-membrane distillation system for produced water treatment. The performance of both the individual forward osmosis (FO) and membrane distillation (MD) systems are firstly studied. The influence of the draw solution and the feed solution temperature are studied respectively for the FO and MD system. In a second part, the FO system is envisaged as a pre-treatment of the MD. Under the tested conditions, the FO and MD permeate fluxes were not constant on time and the authors demonstrated that either the MD or FO could be the limiting step during the time course of the filtration. In my opinion, the results are interesting but some modifications are necessary before acceptation in Membranes.

General comments:

- The authors have decided to plot the permeate flux of the individual FO and MD systems as well as the one of the hybrid FO-MD system against the temperature difference (between the 2 sides of the membrane) (figure 2 and 3). In my opinion, this choice is not wise. As stated by the authors (lines 224-226), the driving force in MD is the difference of vapor pressure between the 2 sides of the membranes and this difference of vapor pressure can be created by a difference of temperature between the feed and permeate side of the MD membrane. However, the vapor pressure variation with temperature is not linear (as it can be calculated using Antoine’s law). Hence, a 20°C difference between the feed an permeate sides will not lead to the same driving force if the hot and cold temperatures are 40 and 20°C or 60 and 40°C respectively. This is well illustrated by figure 3 where an increase in the temperature difference from 20 to 30°C (10°C increase) leads to a much lower increase in permeate flux compared to an increase of the temperature difference from 50 to 60°C (10°C increase). In this study, the cold temperature being constant (and equal to 10°C), it would be much more rigorous and informative to plot the permeate flux against the feed temperature for figure 2 and 3 instead of the temperature difference.

- In figure 3, the permeate flux is plotted against the temperature difference and the data are connected by straight lines. However, as discussed above, in MD the permeate flux can usually be modelled by an exponential curve (Arrhenius type plot).  Accordingly, please remove the straight lines between the data points (for example between the 2 last data point it is highly unlikely that the permeate flux increases linearly).

- Section 3.3, L256-259: the result obtained for Fig 4 are discussed. The decline of the FO permeate flux on time is explained by the dilution of the draw solution due to the different in permeate flux values of MD and FO. I agree that this phenomenon contribute as it lead to the reduction of the osmotic pressure of the draw solution and thus to the one of the FO driving force. However, some others phenomena also occur concurrently: (i) the experiments being carried-out in batch mode, the feed volume lowered with time and thus the concentration of solutes in the feed solution increased accordingly (as high solute rejection is observed, see l. 159-160); (ii) due to the occurrence of reverse salt flux, ions transfer from the draw solution to the feed solution.  These phenomena also tends to lower the osmotic pressure difference between the 2 sides of the FO membranes thus reducing the driving force. Please comment on that. In particular, it would be necessary to indicate and comment the evolution (and finale value) of the volume reduction ratio (VRR) as well as the variation of the feed compartment conductivity.

- We can regret the absence of the data regarding the evolution of the draw solution concentration on time. Thus, as mentioned by the authors, the concentration of the draw solution is a key parameter of the efficiency of the FO process. Here, the draw solution concentration vary on time both due to the dilution effect caused by the difference in permeate flux through the FO and MD membranes and the occurrence of the reverse salt flux. Did the author measure the variation of the draw solution conductivity? This could allow to justify the assumption made of “a low reverse salt flux” (l 295).

-L 297-298: The decrease of the FO flux is attributed to the dynamic of the draw solution and feed solution temperature. Once again, working in batch mode leads to an increasing solute concentration in the feed compartment on time which might contribute to reduce the FO driving force (see comment above). Please comment on the VRR and its variation on time.

- Please comment on the permeate quality during the time-course of the experiment. It is indicated that the MD permeate was analyzed and that the pollutant were fully retained by the membrane system. What about the quality of the draw solution stream? In particular, what is the rejection of the emulsified oil by the FO membrane? Having such information would allow to strengthen the discussion about the membrane wetting issue (l316-323). For the same reason it would also be great to know the type of solutes present in the raw produced water (occurrence of molecules with surfactant properties in addition to oil for example). 

specific comment

- l50: “a total rejection of solute is expected in MD process”. This is true only for non-volatile solutes such as NaCl but not for volatile solutes (such as ethanol-water separation for which MD can be applied). Please modify.

-l 51-555: permeate flux can also drastically increase due to membrane wetting. A confusion seems to be made between fouling and wetting in this sentence. Please adapt.

-Table 1: as mentioned by the authors, the draw solution concentration plays a key role on FO performance. Please indicate the draw solution concentration (or osmotic pressure difference) for the different studies. Please make uniform the number of digits for the permeate flux values.

-section 2.1: please gives more information about the membrane properties. Only the material and membrane geometry/area are provided. Some other parameters are of interest for MD application including membrane pore size, membrane porosity, contact angle values… Please also provide the exact reference of the membrane.

-section 2.2: please correct the title (same as part 2.1); chloride anions usually impair the precision of COD determination if adequate protocol is not used, please comment on that (Was the COD of the draw solution determined?).

- l156: high conductivity of DS is not desired but high osmotic pressure is desired (the driving force is the pressure difference). Please adapt.

- l184-190: the temperature values given as operating conditions are not consistent with the one given in the result section (l244-249), please check 

- figure 1: why is the heater not placed before the MD module? The MD feed passing through the tank placed on the weighting balance, there is a risk of heat loss. Where is the temperature measured (please indicate the location of the temperature sensors on Fig 1)? On Fig 5, it is observed that the temperature difference decreases when the MD flux increases. This could be the consequence of the heat loss due to water vaporization (vaporization latent heat). Knowing the location of the sensors would help to interpret the data.

- l261-262: please indicate the amplitude of the variation (not easy to read on fig 4a). What is the precision? is the variation significant?

typpos:

-l26: “were” instead of “was”

-l84: “why” instead of “because”

-l162: “designed” instead of designated”

-l295: “concentration of the DS” instead of “ concentration the”

-l 328 “importance” instead of “important”

- l331 and l333: “difference” instead of “different”

Round 2

Reviewer 1 Report

The Language usage and paragraph logic have been stated more clearly and the discussion have also been improved. This is suitable for the MDPI

Membranes now. This reviewer suggests to accept.

Reviewer 2 Report

The authors have appropriately replied to most of my comments. In my opinion the paper can be accepted after 2 minor corrections:

 - section 2.1: are you sure of the dense layer thickness (2.3µm)? It seems really thick

- line 360, typo: it should be FS instead of DS
